# Ultra-Fast Vitrification: Minimizing the Toxicity of Cryoprotective Agents and Osmotic Stress in Mouse Oocyte Cryopreservation

**DOI:** 10.3390/ijms25031884

**Published:** 2024-02-04

**Authors:** Jung-Ran Cho, Eun-Hee Yu, Hyun-Joo Lee, In-Hye Kim, Ji-Hye Jeong, Dan-Bi Lee, Seong-Keun Cho, Jong-Kil Joo

**Affiliations:** 1Infertility Center of Pusan National University Hospital, Busan 49241, Republic of Korea; crosslove97@naver.com (J.-R.C.); jjhye@hanmail.net (J.-H.J.); relax8@naver.com (D.-B.L.); 2Laboratory of Animal Reproductive Physiology & Biotechnology, Department of Animal Science, Pusan National University Graduate School, Miryang 50463, Republic of Korea; 3Department of Obstetrics and Gynecology, Pusan National University School of Medicine, Pusan National University Hospital Biomedical Research Institute, Busan 49241, Republic of Korea; hoyyeh@naver.com (E.-H.Y.); atouchofbreeze@gmail.com (H.-J.L.); kihforyou@naver.com (I.-H.K.)

**Keywords:** oocyte cryopreservation, vitrification, ultra-fast vitrification, mitochondria, endoplasmic reticulum, chromosome, meiotic spindle

## Abstract

Globally, women have been adopting oocyte cryopreservation (OC) for fertility preservation for various reasons, such as inevitable gonadotoxic treatment for specific pathologic states and social preferences. While conventional vitrification (C-VIT) has improved the success rate of OC, challenges of possible toxicities of high-concentration cryoprotective agents and osmotic stress persist. To overcome these challenges, we evaluated the ultra-fast vitrification (UF-VIT) method, which reduces the equilibration solution stage exposure time compared to C-VIT by observing mouse oocyte intracellular organelles and embryonic development. Consequently, compared to fresh mouse oocytes, UF-VIT presented significant differences only in endoplasmic reticulum (ER) intensity and mitochondrial (MT) distribution. Meanwhile, C-VIT showed substantial differences in the survival rate, key ER and MT parameters, and embryonic development rate. UF-VIT exhibited considerably fewer negative effects on key MT parameters and resulted in a notably higher blastocyst formation rate than C-VIT. Meiotic spindle (spindle and chromosomes) morphology showed no significant changes between the groups during vitrification/warming (VW), suggesting that VW did not negatively affect the meiotic spindle of the oocytes. In conclusion, UF-VIT seems more effective in OC owing to efficient cytoplasmic water molecule extraction, osmotic stress reduction, and minimization of cell contraction and expansion amplitude, thus compensating for the drawbacks of C-VIT.

## 1. Introduction

Oocyte cryopreservation (OC) was introduced in human-assisted reproductive technology (ART) in the late 1980s [1]. Currently, more women are opting to preserve their oocytes because of delayed marriage and pregnancy plans for social reasons or owing to the unintended consequences of decreased ovarian function caused by gonadotoxic surgery, chemotherapy, or radiotherapy [2,3]. Varlas et al. showed that the use of OC for non-medical reasons has increased from 6% to 20.3% [4]. Accordingly, OC in ART is one of the most actively studied topics and can be a more critical component of this intriguing field of science in the future [5,6].

Compared to the initial adoption of slow freezing for OC, the current widely used method of conventional vitrification (C-VIT) has substantially improved the likelihood of achieving a high survival rate after vitrification/warming (VW) [7,8,9]. However, the results are still unsatisfactory compared to the blastocyst development rate and embryo quality obtained from fresh oocytes, and the discussion on the optimal VW system remains open [10]. C-VIT overcomes the formation of ice crystals within the oocytes, and this is attributable to the characteristics of oocytes, such as a larger cytoplasm, higher water content, and lower surface-to-volume ratio. It is achieved by using high-concentration cryoprotective agents (CPA) and rapid cooling to transform cells into an amorphous glass-like state without ice crystal formation [11,12,13,14]. However, the toxicity of high-concentration CPA, as well as osmotic stress resulting from osmotic equilibrium, needs to be addressed in C-VIT [10,13,15,16]. Physical trauma or high-concentration toxicity of CPA can result in damage to the crucial component of the calcium release system, the endoplasmic reticulum (ER). This triggers the release of calcium ions (Ca^2+^) from the ER, leading to an increase in intracellular calcium ([Ca^2+^]_i_) levels in the mitochondria (MT) and a decrease in mitochondrial membrane potential (ΔΨm) [17,18]. These changes in functionality and structure of cytoplasmic organelles, including the ER and MT, have considerable effects on the survival, fertilization, and developmental potential of the oocytes [12,13,14,19]. 

In terms of the protective effects of CPA, Jin and Mazur emphasized that intracellular water removal by osmosis in the equilibration solution (ES) stage is a crucial protective mechanism, which has a stronger effect than promoting intracellular glass transition [20]. Building on this perspective, Gallardo et al. introduced the concept of “ultra-fast vitrification (UF-VIT)”. This approach involves preserving oocytes with minimal volume just before the introduction of CPA and water, effectively bypassing the typical osmotic equilibrium phase during the ES stage. They successfully achieved rapid vitrification and reported favorable outcomes in both humans and murine models [21].

It should be noted that excessively short exposure times during the ES stage can have a negative effect on the survival and development of vitrified oocytes [22]. Moreover, subsequent studies related to UF-VIT are limited, and its applicability in clinical settings is constrained. Thus, there is a need for studies on human oocytes to generate evidence that could be applied in clinical settings. However, owing to the complex and invasive procedures involved and the associated ethical issues, there are limitations in using human oocytes in scientific research. Therefore, in this study, we considered the need for foundational experiments using a mouse model to confirm the effects of the proposed protocol before its application to human oocytes. Consequently, we utilized a mouse model to investigate whether UF-VIT, which reduces exposure time during the ES stage and alleviates osmotic stress, can have a positive effect on OC. The objective of this study was to examine the survival rates, spindle positioning, ER distribution and function, MT distribution and function, and embryonic development in mouse oocytes to determine whether UF-VIT is superior to C-VIT. 

## 2. Results

### 2.1. Survival Rates of Oocytes

Oocyte survival rates were analyzed in comparison to the control oocytes (200/200, set as 100%). Significant differences were observed in only the C-VIT-treated oocytes (200/210, *p* < 0.05), whereas the UF-VIT-treated oocytes (200/203, *p* = 0.745) showed no significant difference. A comparison of the survival rates between the C-VIT and UF-VIT (*p* = 0.263) groups showed no significant difference.

### 2.2. Distribution and Fluorescence Intensity of ER

The distribution of ER was analyzed according to the criteria shown in Figure 1A,B, and the distribution in the equatorial part was classified as even throughout the perispindle and the entire oocytes (considered normal) or as small areas of damaged ER (considered abnormal). In the cortical region, analysis was based on the presence (considered normal) or absence (considered abnormal) of a cluster containing inositol 1,4,5-trisphosphate (IP_3_) receptor (IP_3_R1) [23].

In the equatorial region, the analysis of ER distribution showed a significant difference in the percentage of normal distribution between the control (46/50, 92%) and C-VIT-treated oocytes (33/50, 66%, *p* < 0.01). However, there was no significant difference between UF-VIT-treated oocytes (42/50, 84%, *p* = 1.000) and the control (Figure 1A,C). Additionally, analysis of ER distribution in the cortical part, based on the presence of clusters, showed a significant difference between the control (44/50, 88%) and C-VIT-treated oocytes (27/50, 54%, *p* < 0.01). However, no significant difference was observed between UF-VIT-treated oocytes (38/50, 76%, *p* = 0.576) and the control (Figure 1B,D). No significant differences were found between the vitrification methods in the equatorial (*p* = 0.190) and cortical (*p* = 0.106) portions.

The mean fluorescence intensity of ER differed significantly between the control group (189 ± 26.49 gray) and both C-VIT-treated (169 ± 25.47 gray, *p* < 0.001) and UF-VIT-treated oocytes (174 ± 19.01 gray, *p* < 0.01) (Figure 1E). However, there was no significant difference in the fluorescence intensity between the C-VIT- and UF-VIT-treated oocytes (*p* = 0.884).

### 2.3. Distribution and Fluorescence Intensity of MT and Mitochondrial Membrane Potential Analysis 

The distribution of MT was classified as either evenly distributed throughout the entire oocyte (considered normal) or aggregated in damaged areas (considered abnormal) (Figure 2A) [16,24].

Analysis of MT distribution revealed significant differences between the control (43/50, 86%) and both C-VIT- (7/50, 14%, *p* < 0.001) and UF-VIT-treated oocytes (23/50, 46%, *p* < 0.001). C-VIT-treated oocytes exhibited significantly greater MT aggregation than UF-VIT-treated oocytes (*p* < 0.01) (Figure 2C).

There was a significant difference in ΔΨm between the C-VIT-treated (red/green; 98.14/162.44 gray, 0.61, *p* < 0.001) and control oocytes (109.85/140.38 gray, 0.80), but no significant difference was observed between UF-VIT-treated oocytes (107.76/138.97 gray, 0.79, *p* = 1.000) and the control. C-VIT-treated oocytes had significantly lower ΔΨm values than UF-VIT-treated oocytes (*p* < 0.001) (Figure 2B,D).

There was a significant difference in the mean fluorescence intensity of MT between C-VIT-treated oocytes (148 ± 24.64 gray, *p* < 0.001) and the control (174 ± 11.73 gray). However, no significant difference was observed between UF-VIT-treated oocytes (173 ± 22.29 gray, *p* = 1.000) and the control. C-VIT-treated oocytes exhibited adverse effects compared to UF-VIT-treated oocytes (*p* < 0.001) (Figure 2E).

### 2.4. Meiotic Spindle in and Chromosomal Morphology of Oocytes

After 2–3 h of warming, the use of both C-VIT (98%, *p* = 1.000) and UF-VIT (100%, *p* = 1.000) cryopreservation methods resulted in no significant difference in the recovery of mouse oocyte meiotic spindle and chromosomes, with both methods achieving complete recovery, similar to that in fresh mouse control oocytes (100%) (Figure 3A,B).

### 2.5. Comparison of Oocyte Volume Excursion Based on CPA Exposure Time

Both protocols induced rapid dehydration of the oocytes, reducing their isotonic volume to approximately 48%–50% after 30 s of exposure to the ES stage. In the case of UF-VIT, CPA penetration was initiated after 30 s of ES exposure, leading to a slight re-expansion of the volume to about 57%, followed by contraction to 49% in the VS stage. On the contrary, in C-VIT, the exposure time during the ES stage was extended to 10 min to achieve equilibration, resulting in a substantial re-expansion in volume, reaching up to approximately 94%. Following exposure to the VS stage, the volume decreased to 71% of the isotonic volume. These differences in volume excursion clearly indicate that oocytes frozen with C-VIT undergo more marked changes than those frozen with UF-VIT (Figure 4A–C).

### 2.6. Comparing Embryonic Development Rates

Compared to the control group, C-VIT showed significant differences in the rates of fertilization (92.5% vs. 74.3%, *p* < 0.05), cleavage (82.5% vs. 53.8%, *p* < 0.01), and blastocyst formation (55.0% vs. 25.6%, *p* < 0.01). However, UF-VIT did not exhibit significant differences compared to the control group (fertilization rate; 80.0%, *p* = 0.105, cleavage rate; 67.5%, *p* = 0.121, blastocyst formation rate; 47.5%, *p* = 0.502). When comparing the two methods, a significant difference was observed in the blastocyst formation rate (25.6% vs. 47.5%, *p* < 0.5) (Figure 5A–C). 

## 3. Discussion

The research results indicate that the UF-VIT group showed no significant differences compared to the control group, except in terms of ER function (fluorescence intensity) and MT distribution. UF-VIT had fewer negative effects on all parameters related to MT and showed a superior blastocyst formation rate than C-VIT. However, the C-VIT group showed negative effects on survival rate and all parameters related to ER and MT, as well as embryonic development, compared to the control group. Through this, we propose UF-VIT as a solution to address the challenges faced by C-VIT, aiming to reduce cell toxicity caused by high-concentration CPA exposure and osmotic stress owing to volumetric changes. The interpretation of this evidence is as follows.

Firstly, the study demonstrated that removing intracellular water is more crucial than the protective effect of CPA penetration into oocytes [20]. Since all CPAs have potential toxicity, prolonged exposure to high-concentration CPA negatively affects the function of intracellular organelles in oocytes [6]. In particular, the observed function of MT in this study plays a crucial role in supplying ATP and supporting embryonic development [25,26]. ΔΨm serves as an important indicator of cell damage, reflecting the ratio of red to green fluorescence intensity [2,3,27]. However, reduced MT function due to CPA toxicity leads to energy depletion, and the loss of ΔΨm can result in the release of proteins, such as cytochrome C, into the intermembrane space, leading to cell death or decreased energy supply efficiency [28,29]. Therefore, the impairment of MT function and ΔΨm in C-VIT, where equilibrium is required during the ES stage, suggests a lower developmental potential compared to the control group, confirming the results of lower blastocyst development.

However, the ER function, a crucial component of the Ca^2+^-release system that increases [Ca^2+^]_i_ levels during fertilization, showed significant damage in both C-VIT and UF-VIT [9,17,23]. This can be explained based on a study suggesting that penetrating CPA, particularly dimethyl sulfoxide (DMSO), can induce Ca^2+^ release from the ER, resulting in a decrease in ER Ca^2+^ levels and an abnormal increase in cytoplasmic [Ca^2+^]_i_ levels [17]. Furthermore, this underscores the necessity for additional research into penetrating CPA that can serve as an alternative to DMSO.

Secondly, during the freezing process, a significant change in the volume of oocytes can lead to “osmotic shock” occurs, which can affect the distribution of oocyte organelles. To address this issue, it is important to limit the magnitude of cell volume changes caused by osmotic stress [15,30]. During the ES stage in C-VIT, water molecules are expelled from the cells owing to osmotic pressure, thereby resulting in a minimal volume. Subsequently, with the influx of CPA and water, an osmotic equilibrium is gradually achieved. During the VS stage, water is further eliminated, resulting in glass formation. On the contrary, UF-VIT involves the formation of a minimal volume due to osmotic pressure, followed by immediate vitrification, resulting in a contraction–contraction process, as opposed to the contraction–expansion–contraction process observed in C-VIT [30]. The equatorial part of the ER observed in this study is involved in forming a network around the spindle or entire membrane channel of the oocyte to regulate the [Ca^2+^]_i_ levels. In addition, clusters present in the cortical part include inositol 1,4,5-trisphosphate (IP_3_) receptors (IP_3_R1), which are activated by sperm entry and play a role in [Ca^2+^]_i_ release and periodic calcium oscillation, thus contributing to oocyte activation [15,31,32,33,34]. Therefore, the UF-VIT group, with minimal volume changes, exhibited a similar ER distribution pattern to the control group and did not experience negative effects on embryonic development, unlike C-VIT.

However, both UF-VIT and C-VIT had negative effects on the distribution of MT. Despite reducing volume changes, the contraction process, which is necessary to eliminate intracellular moisture, led to MT damage. Nevertheless, compared to C-VIT, UF-VIT resulted in less damage to MT distribution owing to smaller volume excursion changes, yet further research is needed to explore methods to alleviate instantaneous contraction. 

Additionally, one of the main reasons for the low clinical efficacy of cryopreservation of human oocytes at the metaphase II (MII) stage is the loss of spindle, which are sensitive to CPA and low temperatures, and their disassembly [1,11,35,36,37,38]. However, although depolymerization occurs during freezing, almost complete repolymerization occurs after warming and subsequent culture at 37 °C for 2–3 h [3,14]. Furthermore, there is also research indicating that the spindle does not disappear at any stage during the freezing process [39]. Therefore, if the meiotic spindle and chromosomes in oocytes maintain their normal morphology after warming, intermediate position changes should not cause problems in normal embryonic development [27]. In this study, 2D confocal microscopy images revealed abnormal morphology of the meiotic spindle and chromosomes depending on their locations. However, 3D images showed that C-VIT and UF-VIT retained over 98% of the normal morphology of the meiotic spindle and chromosomes. These findings suggest that the occurrence of aneuploidy in the meiotic spindle and chromosomes in VW oocytes increases with advancing age, indicating that it is not caused by CPA toxicity [16]. Therefore, we can understand that the difficulties in the development from VW oocytes to the blastocyst stage primarily stem from the failure of intracellular organelles, including ER and MT, to function properly in the required location rather than issues related to the meiotic spindle and chromosomes.

According to the research findings, the C-VIT method may cause potential damage to crucial cell organelles in mouse oocytes due to CPA toxicity and osmotic stress. In contrast, UF-VIT showed relatively minimal damages, likely attributed to the reduced CPA exposure time during the ES stage. UF-VIT validated this by visually assessing the organelle status within oocytes and evaluating embryonic development rates. However, additional research is needed to enhance ER intensity and MT distribution in UF-VIT, and additional clinical studies using human oocytes are essential before applying UF-VIT, considering significant differences between mouse and human oocytes. Lastly, this study introduces a novel approach to cryopreserving oocytes and sheds light on addressing potential issues, emphasizing the need for further practical applications and clinical investigations.

## 4. Materials and Methods

### 4.1. Animals

All procedures involving mice were conducted in accordance with the guidelines and regulations provided by the Pusan National University Hospital Institutional Animal Care and Use Committee (Approval No. PNUH-2022-202) for the protection and ethical use of experimental animals. All experiments were approved by the Institutional Animal Care and Use Committee of Pusan National University Hospital under the same approval number. Furthermore, all methods were performed in accordance with the ARRIVE guidelines.

The mice were maintained in a controlled environment, with regulated light–dark cycles (12 h cycles) and a constant temperature (24 °C ± 2 °C). They were fed a standard diet and received care from the Experimental Animal Department at Pusan National University Hospital. The mice were anaesthetised using carbon dioxide (CO_2_). Mature oocytes were collected from 7-week-old female B6D2F1 mice (Hana Corporation, Busan, Republic of Korea) and randomly assigned to three groups (*n* = 240 oocytes/group).

### 4.2. Oocyte Collection

To collect MII mouse oocytes, 10 IU of pregnant mare serum gonadotropin (Daesung, Gyeonggi, Republic of Korea) was administered intraperitoneally, followed by the administration of 10 IU of human chorionic gonadotropin (hCG; Daesung) after 48 h to induce superovulation. Oocytes were collected from the ampulla of the oviduct 15–16 h post-hCG injection. Cumulus–oocyte complexes were treated with 80 IU of hyaluronidase to remove cumulus cells. Denuded oocytes were washed twice in M2 medium (Sigma-Aldrich, St. Louis, MO, USA) and transferred to M16 medium (Sigma-Aldrich). Mature oocytes were classified as normal or abnormal based on their morphological characteristics and were assessed using a stereomicroscope [16]. In total, 740 oocytes were utilized, with 9 oocytes allocated to each vitrification method for measuring relative oocyte volume and 40–50 oocytes allocated to each experimental protocol. The control group consisted of fresh MII mouse oocytes.

### 4.3. Vitrification and Warming Protocols for C-VIT and UF-VIT

The solution used for vitrifying mouse MII oocytes is composed of permeable CPA (ethylene glycol, EG (Sigma-Aldrich) and DMSO (Sigma-Aldrich)) and a non-permeable CPA (sucrose, Sigma-Aldrich). The C-VIT protocol, which is currently the standard procedure, involves exposing oocytes to ES stage (7.5% *v*/*v* EG + 7.5% *v*/*v* DMSO + 20% serum substitute supplement (SSS) + Dulbecco’s phosphate-buffered saline (DPBS)) for 10 min. Subsequently, they are exposed to VS stage (15% *v*/*v* EG + 15% *v*/*v* DMSO + 0.5M sucrose + 20% SSS + DPBS) for 1 min. However, the UF-VIT protocol introduced by Gallardo et al. exposes oocytes to the ES and VS stages for 1 min each.

In accordance with the C-VIT and UF-VIT protocols, vitrified oocytes were loaded onto a closed vitrification carrier (Reprocarrier; Moduscience, Sejong, Republic of Korea) and rapidly immersed in liquid nitrogen. Loading, closing, and cooling of the vitrification device were completed within the 1 min duration of the VS stage, with no further processing required thereafter.

To warm the vitrified oocytes, 1 M of sucrose solution (1 M of sucrose + 20% SSS + DPBS) was pre-incubated for 90 min. The vitrified oocytes were sequentially exposed to 1 M sucrose solution at 37 °C for 1 min, 0.5 M of sucrose solution (0.5 M of sucrose + 20% SSS + DPBS) at room temperature (RT, 25–26 °C) for 3 min, base media 1 (20% SSS + DPBS) at RT for 5 min, and base media 2 (20% SSS + DPBS) at RT for 1 min. The volume of each warming solution was 1 mL. Thawed oocytes were washed twice in M16 medium and cultured at 37 °C with 5% CO_2_ to facilitate recovery. Additionally, studies by Chen. S and Amidi. F et al. have reported that the organelles, including the spindle and mitochondria, recover within 120–180 min post-thawing. Therefore, cultivation was completed 150 min after thawing, and all observations were conducted accordingly (Figure 6) [3,15,21].

### 4.4. Survival Rates of Oocytes

The survival rate of oocytes was compared 150 min after warming. Degeneration was considered if cytoplasmic changes or fragmentations were evident. The survival rate (number of surviving oocytes/warmed oocytes) was evaluated and compared between experimental groups. The surviving oocytes were used in further experiments.

### 4.5. Live Imaging of ER and MT in Mouse Oocytes Using a Confocal Microscope

To investigate the distribution and fluorescence intensity of the ER and MT, MII mouse oocytes vitrified and warmed using C-VIT and UF-VIT were utilized. After 150 min of warming, the oocytes were cultured in ER-Tracker™ Green (200 nmol/L, E34251; Invitrogen, Carlsbad, CA, USA) or MitoTracker™ Red (100 nmol/L; Invitrogen, Carlsbad, CA, USA) at 5% CO_2_ for 20 min at 37 °C. They were then mounted on a slide, and live images of the ER (green) and MT (red) of the oocytes were directly obtained using a laser-scanning confocal microscope (LSCM; TCS SP8; Leica Microsystems, Wetzlar, Germany). Fluorescence intensity analysis was performed using LAS X software (v3.5.1, Leica).

### 4.6. Measurement of MT Membrane Potential Using JC-1 Staining

MII mouse oocytes were prepared for the assessment of ΔΨm with 2 µM of JC-1 (5,50,6,60-tetrachloro-1,10,3,30-imidacarbocyanine iodide, T3168; Molecular probes, Life Technologies, Carlsbad, CA, USA) and incubated at 5% CO_2_ for 10 min at 37 °C [40]. Oocytes were analyzed using an LSCM, with fluorescein isothiocyanate (green) and rhodamine isothiocyanate (red) channels and the LAS X software v3.5.1, which allows the quantitative measurement of fluorescence signal intensities of JC-1 monomers (relatively low potential, <100 mv, green) and JC-1 aggregates (high potential, >140 mv, red) [14,41]. ΔΨm was calculated as the ratio of red to green for each oocyte.

### 4.7. Immunofluorescence Staining of Oocytes 

To evaluate the morphology of the meiotic spindle using α-tubulin in microtubules (a spindle marker) and chromosomes using 6-diamidino-2-phenylindole (DAPI) immunofluorescence staining of mature oocytes, the oocytes were fixed in phosphate-buffered saline (PBS) containing 4% paraformaldehyde for 30 min at RT. The fixed oocytes were washed three times for 5 min each in 0.1% polyvinyl alcohol (PVA) in PBS and then permeabilized in 0.1% Triton X-100 in PBS for 20 min at RT. Blocking was performed in 3% bovine serum albumin in 0.1% PVA/PBS for 1 h at RT, followed by incubation with an Alexa Fluor 488 conjugated α-tubulin monoclonal antibody (1:100, 322,588; Invitrogen) at RT overnight. After washing with 0.05% tween 20 in PBS three times for 10 min each, the oocytes were mounted on a slide with a mounting medium containing DAPI (H-1200, Vector Laboratories, Burlingame, CA, USA) and observed under an LSCM. The results were analyzed using the LAS X software v3.5.1 after the green color was converted to false color (red) for better visualization. In the meiotic spindle, the microtubules (red) were organized in a barrel shape, and only those with well-aligned chromosomes (blue) in the center were considered normal. In cases where only cross-sections were visible depending on their location, the meiotic spindles were analyzed by observing them in 3D for further analysis [16]. 

### 4.8. Image Acquisition and Relative Volume Excursion of Oocytes

To measure the relative volume excursion of oocytes during the vitrification process, it was assumed that the oocytes maintained a perfectly spherical shape during both contraction and expansion. Images were captured using video recording (Olympus IX53, Tokyo, Japan) at a magnification of 100×. Initially, oocytes were loaded into a 2 µL base medium drop and held with a holding pipette, and then 300 µL of ES was added to initiate the osmotic process. After the exposure time to ES, ES was removed, and 300 µL of VS was added. For the first 60 s in both ES and VS, video frames were extracted every 10 s. In the case of ES of C-VIT, after the initial 60 s, video frames were extracted at 60 s intervals for analysis, determining the volumetric excursion of the oocytes relative to the initial isotonic volume. Still images of the oocytes were analyzed using Image J software v1.53 to determine their area. To minimize distortion caused by irregularities in the spherical shape of the oocytes during contraction and expansion, the radius was calculated from the oocytes’ bidimensional area. Through this, the total volume of the oocytes was determined, assuming a perfect spherical shape maintained throughout the process [21,42]. 

### 4.9. Intracytoplasmic Sperm Injection

We collected sperm by performing cauda epididymis dissection from 8- to 9-week-old male B6D2F1 mice. The retrieved sperm were immediately incubated in pre-warmed Human Tubal Fluid media (HTF; FUJIFILM Irvine Scientific, San Diego, CA, USA) supplemented with 10% SSS for 1 h to enhance sperm capacitation. Before performing ICSI, to accommodate the insertion of the injection pipette due to the small size of mouse oocytes, we fabricated a holding pipette with a trumpet-shaped opening and an inner diameter of approximately 45–55 µm using a micro forge (MF-900, Narishige, Tokyo, Japan) as described by Lyu, Q. F et al. [43]. Subsequently, the cytoplasm of mouse oocytes was aspirated into the fabricated pipette, and ICSI was performed using a commercial injection pipette (K-MPIP-3130, Cook Medical, Bloomington, IN, USA) under equipment fitted with an inverted microscope and micromanipulation system (TE300, Nikon, Tokyo, Japan). They were then cultured in M16 medium following ICSI. We evaluated the fertilization rates (number of 2-cell embryos/survived oocytes), cleavage rate (number of 4–8 cell embryos/survived oocytes), and blastocyst formation rate (number of blastocysts/survived oocytes) between the experimental groups [16].

### 4.10. Statistical Analysis

Statistical analyses were performed using the language R (http://cran.r-project.org (accessed on 15 December 2022)) version 4.1.3. Analyses were performed using a one-way analysis of variance followed by post hoc analysis. For continuous data, independent *t*-tests or Wilcoxon rank sum tests were used, whereas for categorical data, the Chi-squared test or Fisher’s exact test was employed. Continuous data are presented as mean ± standard deviation, whereas categorical data are expressed as *n* (%). Statistical significance was set at *p* < 0.05. 

## Figures and Tables

**Figure 1 ijms-25-01884-f001:**
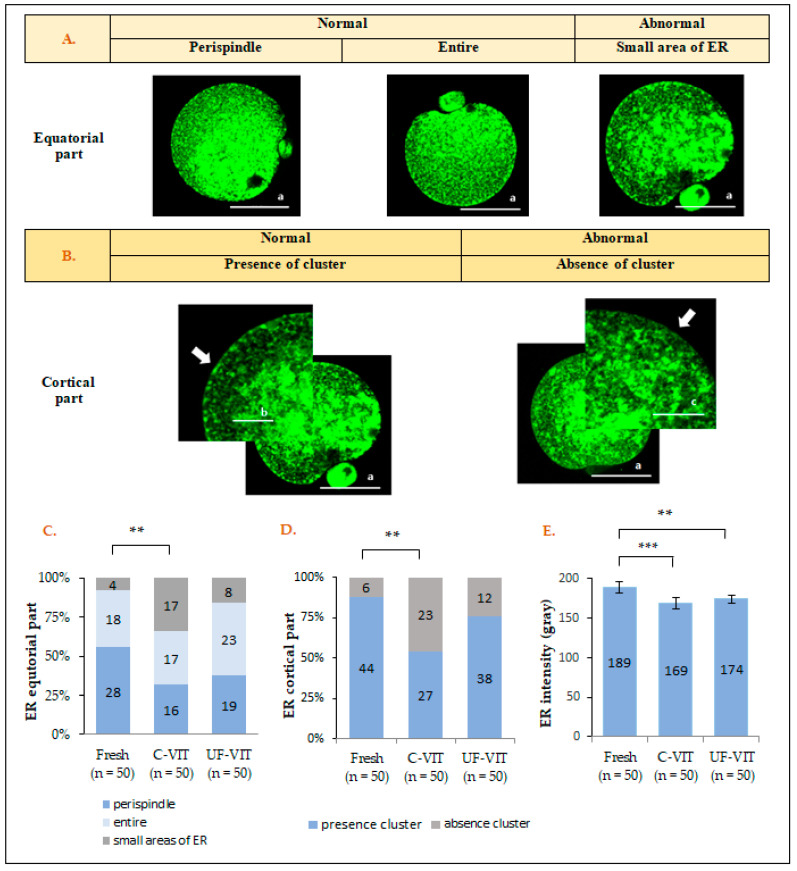
Fluorescent live cell images showing the fluorescence intensity and distribution pattern of oocyte ER and their analysis. (**A**) Live fluorescent image showing the distribution of normal and abnormal regions in the equatorial part; (**B**) Image magnifications at 2.6x (**left**) and 3x (**right**) for assessing the presence of cluster in the cortical part (white arrows); (**C**,**D**) analysis of ER equatorial and cortical part distribution; (**E**) analysis of mean ER fluorescence intensity. Values are presented as mean ± SEM. Scale bar a = 58 μm, b = 22.3 μm, c = 19.4 μm. ** *p* < 0.01, *** *p* < 0.001.

**Figure 2 ijms-25-01884-f002:**
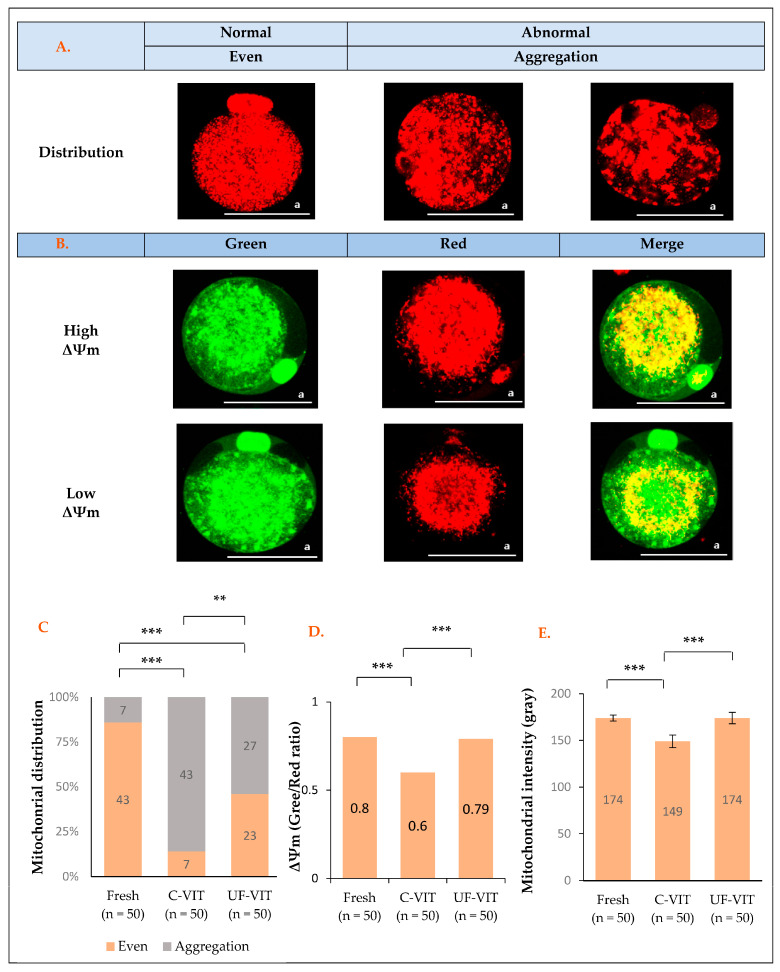
Fluorescent live cell images showing fluorescence distribution pattern and intensity of oocyte MT and their analysis. (**A**) Live fluorescent images of MT distribution; (**B**) Live fluorescent images of MT membrane potential; (**C**–**E**) Analysis of mean MT distribution, ΔΨm, and MT intensity. Values are presented as mean ± SEM. Scale bar a = 58 μm. ** *p* < 0.01, *** *p* < 0.001.

**Figure 3 ijms-25-01884-f003:**
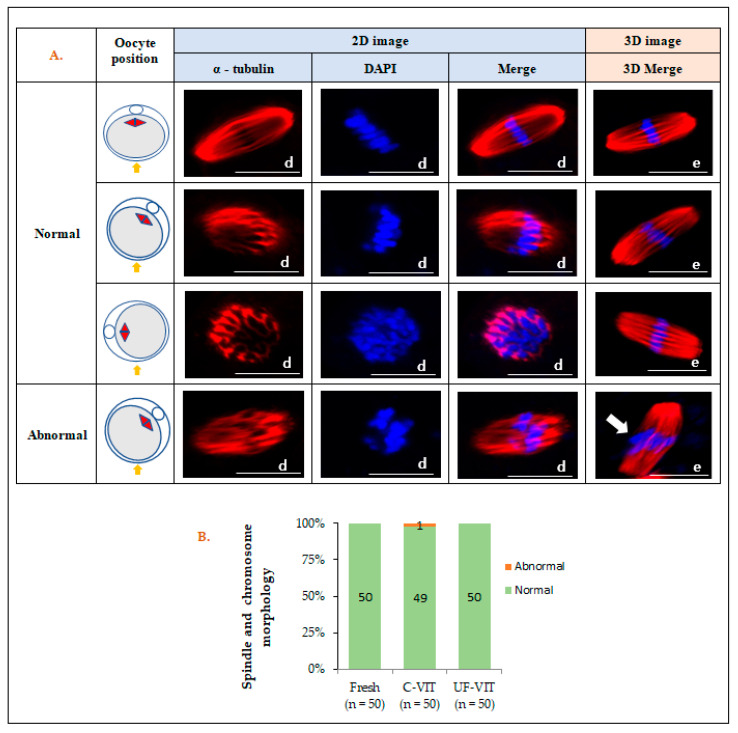
2D and 3D laser-scanning confocal microscopy images of the meiotic spindle (red) and chromosomes (blue) in metaphase II (MII) mouse oocytes. (**A**) Morphological appearance of the meiotic spindle (α-tubulin) and chromosomes (DAPI) varies depending on the position of the oocyte in 2D images and in 3D observations (yellow arrow: microscope observation direction, white arrow: abnormal chromosomes morphology). (**B**) Analysis of meiotic spindle and chromosomes morphology. Scale bar d = 19.3 µm, e = 20 µm.

**Figure 4 ijms-25-01884-f004:**
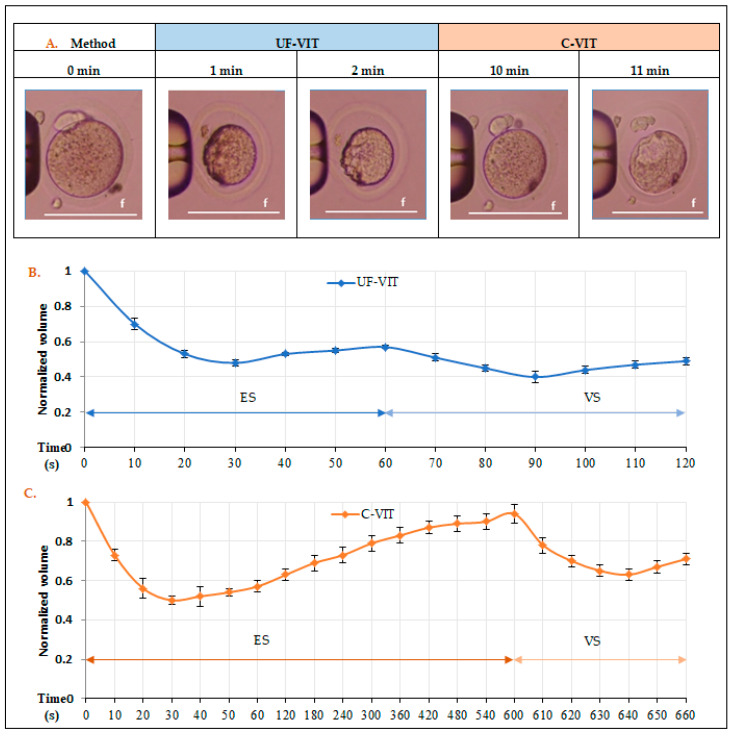
Recorded results of volumetric excursion in mouse MII oocytes according to C-VIT and UF-VIT freezing processes. (**A**) Changes in the morphology of mouse oocytes upon exposure to ES and VS; (**B**,**C**) A graph depicting the relative volume excursion of oocyte over time, initiated by ES exposure at *t* = 0 according to the protocols of UF-VIT and C-VIT. The graphs represent mean values and standard error bars at each time point. Scale bar f = 100 µm.

**Figure 5 ijms-25-01884-f005:**
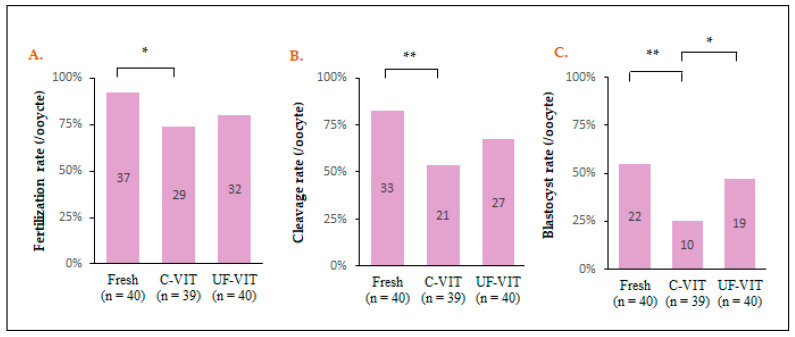
Comparison of embryonic development after V/W with C-VIT and UF-VIT. (**A**–**C**) The bar graphs show the fertilization, cleavage, and blastocyst formation rates of oocytes following intracytoplasmic sperm injection (ICSI). * *p* < 0.05, ** *p* < 0.01.

**Figure 6 ijms-25-01884-f006:**
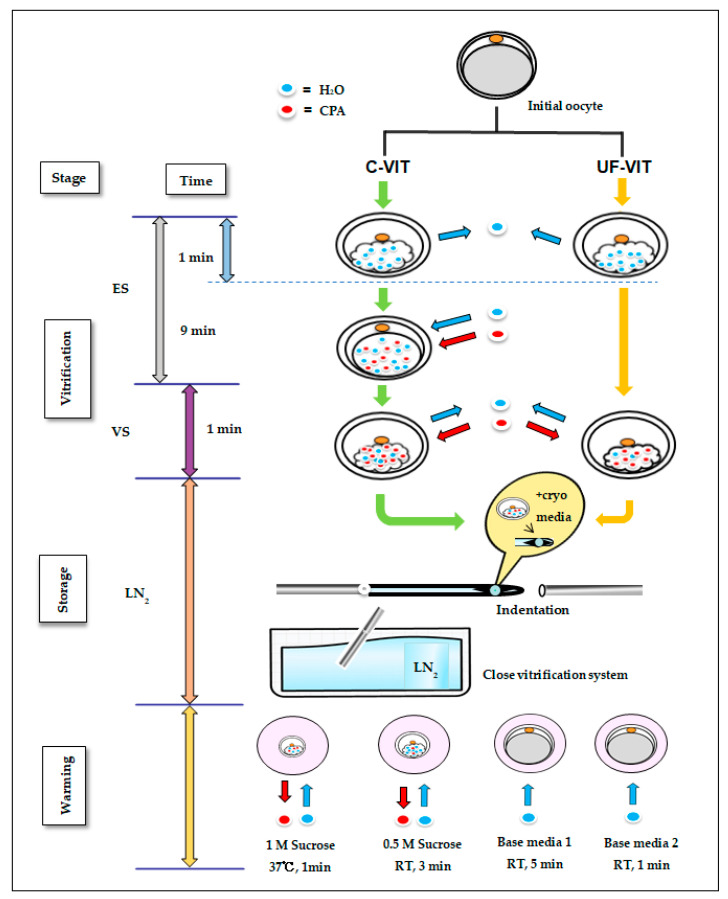
Vitrification and warming processes of C-VIT and UF-VIT. Schematic representation of the vitrification and warming processes of C-VIT and UF-VIT. The blue arrows represent the movement of H_2_O, while the red arrows represent the movement of CPA.

## Data Availability

The datasets generated during and/or analyzed during the current study are available from the corresponding author upon reasonable request.

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
