# Peer review of "Ultra-Fast Vitrification: Minimizing the Toxicity of Cryoprotective Agents and Osmotic Stress in Mouse Oocyte Cryopreservation"

_ijms, 2024, doi:10.3390/ijms25031884_

Round 1

Reviewer 1 Report

Comments and Suggestions for Authors

The aim of the study was to investigate the effects of the ultra-fast vitrification (UF-VIT) method, which reduces the equilibration solution stage exposure time compared to conventional vitrification (C-VIT), by observing mouse oocyte intracellular organelles and embryonic development.

In general, the main objectives have been achieved and sufficiently supported by experimental data, however, the authors should clarify some methodological procedures.

Minor comments:

Materials  and Methods

·         Pag.10, line 318-320: please, the authors should specify the composition of solution sucrose 1 M and sucrose 0.5 M.

·         Pag.10, line 319-320: please, the authors should specify the composition of media base 1 and media base 2.

·         Pag.12, line 334: the authors describe a warming time of 150 min before to perform the analyses of oocyte. Is the duration of this post warming time standardized? Do the authors have experience on the effect of the duration of time post warming on oocyte quality in mouse? It would be better specify why this time (150 min) is chosen for the analysis of oocyte with the support of references.

Results

In Figures 1,2,3, 4 and 5 the letter are uppercase (A, B, C D), while in the legends and in the text the letters are lowercase. This is unclear. Please standardize text, legend and figures with all capital letters.

Reviewer 2 Report

Comments and Suggestions for Authors

Comments about the manuscript:

“Ultra-fast vitrification: Minimizing the toxicity of cryoprotective agents and osmotic stress in mouse oocyte cryopreservation”

Oocyte cryopreservation (OC) is now used in various pathological cases. However, frozen oocytes may be affected by the toxicity of cryoprotective agents and osmotic stress. The aim of the study presented here was the evaluation of an ultra-rapid vitrification method (UF-VIT), the aim of which is to reduce the time the oocyte is exposed to these disadvantages. To do this, the authors observed the intracellular organelles and the development of mouse oocytes subjected to such ultra-rapid vitrification compared to fresh oocytes or conventionally frozen oocytes. Differences were observed. Ultra-rapid vitrification has fewer negative effects with a higher blastocyst formation rate than the conventional method. The results obtained are encouraging.

This work, which seems to me well done and well written, is clear despite the large number of observations. I will only make a few minor remarks.

General: as is often the case in this type of work, the number of abbreviations is large and the reader can get a little lost. I suggest that a glossary with all the abbreviations be given at the end of the text, this could make the document easier to read.

Page 3, figure 1. “Scale bar = 58 μm”: I did not find scale bars on the pictures concerned.

Page 5, figure 2. The same: I did not find scale bars on pictures.

Page 6, figure 3. The same: “2D scale bar = 19.3 µm, 3D scale bar = 256.2 m”: I did not find scale bars on pictures. I suppose it is 256;2 µm instead of 256;2 m.

Page 7, figure 4: add a scale bar on the micrographs.

Page 13. “with a micromanipulation system (TE300, Nikon, Japan), as previously described [43]”: for a scientific article, it would be useful to briefly describe the technical aspects.
